# Diaphragmatic Activation Correlated with Lumbar Multifidus Muscles and Thoracolumbar Fascia by B-Mode and M-Mode Ultrasonography in Subjects with and without Non-Specific Low Back Pain: A Pilot Study

**DOI:** 10.3390/medicina59020315

**Published:** 2023-02-08

**Authors:** Alicia Martin Perez, Samuel Fernández-Carnero, Cristina Sicilia-Gomez-de-Parada, Nicolas Cuenca-Zaldívar, Fermin Naranjo-Cinto, Daniel Pecos-Martín, Tomás Gallego-Izquierdo, Susana Nuñez-Nagy

**Affiliations:** 1Grupo de Investigación en Fisioterapia y Dolor, Departamento de Fisioterapia, Facultad de Enfermería y Fisioterapia, Universidad de Alcalá, 28801 Alcalá de Henares, Spain; 2Research Group in Nursing and Health Care, Puerta de Hierro Health Research Institute—Segovia de Arana (IDIPHISA), 28222 Madrid, Spain

**Keywords:** non-specific low back pain, diaphragm, lumbar multifidus muscles, thoracolumbar fascia, rehabilitative ultrasound imaging

## Abstract

*Background and Objectives*: The diaphragm, the lumbar multifidus muscles, and the thoracolumbar fascia (TLF) execute an important role in the stability of the lumbar spine and their morphology has been modified in subjects with non-specific low back pain (NS-LBP). While it is true that three structures correlate anatomically, the possible functional correlation between them has not been investigated previously in healthy subjects nor in subjects with NS-LBP. The aim of the present study was to examine this functional nexus by means of a comparison based on ultrasonographic parameters of the diaphragm, the lumbar multifidus muscles, and the TLF in subjects with and without NS-LBP. *Materials and Methods*: A sample of 54 (23 NS-LBP and 31 healthy) subjects were included in the study. The thickness and diaphragmatic excursion at tidal volume (TV) and force volume (FV), the lumbar multifidus muscles thickness at contraction and at rest, and the TLF thickness were evaluated using rehabilitative ultrasound imaging (RUSI) by B-mode and M-mode ultrasonography. The diaphragm thickening capacity was also calculated by thickening fraction (TF) at tidal volume and force volume. *Results:* There were no significant differences recorded between the activation of the diaphragm and the activation of the lumbar multifidus muscles and TLF for each variable, within both groups. However, there were significant differences recorded between both groups in diaphragm thickness and diaphragm thickening capacity at tidal volume and force volume. *Conclusions:* Diaphragmatic activation had no functional correlation with the activation of lumbar multifidus muscles and TLF for both groups. Nevertheless, subjects with NS-LBP showed a reduced diaphragm thickness and a lower diaphragm thickening capacity at tidal volume and force volume, compared to healthy subjects.

## 1. Introduction

Non-specific low back pain (NS-LBP) is defined as a pain focused on lumbar region that it produces symptoms of unknown cause. In approximately 10–15% of subjects with NS-LBP, it becomes chronic pain and it affects men and women indistinctly [1]. Although the pathophysiological mechanism that leads to the appearance of NS-LBP is not fully defined, scientific evidence has shown that patients with NS-LBP develop changes in musculoskeletal and connective structures related to the maintenance of the stability of the lumbar spine.

The diaphragm is one of the muscles that contributes to the stabilization of the lumbar spine through the intra-abdominal pressure (IAP) [2]. The diaphragm raises the IAP prior to the start of a functional task with the upper limbs and keeps it at high levels while the task is being carried out [3]. The increase in IAP is also achieved owing to the activation of the transversus abdominis [4]. However, the diaphragm must develop its respiratory function at the same time. Consequently, the increase in demand for respiratory function reduces the ability to perform the stability function, and vice versa [5,6]. Several studies have shown that subjects with low back pain have more altered breathing patterns [7], as well as a lower thickness [8] and diaphragmatic excursion [6,9] compared to healthy subjects. Scientific evidence has also documented an asymmetric recruitment of portions of the diaphragm in individuals with chronic low back pain [10]. Despite these investigations, at present, no standard has been established for diaphragmatic activation in individuals with NS-LBP.

The thoracolumbar fascia (TLF) is another of the structures that contributes to the stability of the lumbar spine [11]. The transversus abdominis (TrA), the internal oblique (IO), and the external oblique (EO) are inserted on the middle layer of this TLF [12]. The force generated from the increase in IAP and the contraction of the abdominal muscles puts this connective tissue in tension which helps to stabilize the lumbar spine [13]. It has been documented a thickness greater than 25% in the TLF and a greater disorganization of the connective tissue layers that compose the TLF in subjects with chronic low back pain [14].

The lumbar multifidus muscles are also stabilizers of the lumbar spine [15]. In addition, the posterior layer of the TLF envelops the lumbar multifidus muscles [16] and the contraction of this musculature produces an increase in the length of the TLF [17]. In subjects with unilateral low back pain, it has been documented a bilateral asymmetry of the lumbar multifidus muscles between the symptomatic and asymptomatic side [18]. In addition, in cases of acute low back pain, an atrophy of the ipsilateral lumbar multifidus muscles next to pain has been demonstrated [19].

For the analysis of these muscular and connective structures, the techniques traditionally employed are based on imaging tests, including radiography, ultrasonography, fluoroscopy, magnetic resonance imaging, and electromyographic tests [20]. Ultrasonography has advantages including its noninvasive technique, nonionizing radiation [21], reproducibility, well tolerated by the patient, and has good inter- and intra- observer reliability [22,23]. Specifically, rehabilitative ultrasound imaging (RUSI) is considered a method of choice for the evaluation of musculoskeletal structures by physiotherapists [24,25].

Therefore, the diaphragm, the TLF and the lumbar multifidus muscles are anatomically correlated. We presume that there is also a functional correlation between diaphragmatic activity and the activity of the lumbar multifidus muscles and the TLF in healthy subjects. In addition, it is suspected that in the presence of NS-LBP, the relationship between these structures is modified. However, no study has evaluated the possible functional correlation between these structures.

For all the above reasons, the purpose of this study was to perform a comparison based on the measurement of ultrasonographic parameters of the diaphragm, the lumbar multifidus muscles, and the TLF in subjects with and without NS-LBP.

## 2. Materials and Methods

### 2.1. Study Design and Ethical Statement

A pilot study was conducted according to the Guidelines for Reporting Reliability and Agreement Studies (GRASS) [26]. This study was executed at the Faculty of Nursing and Physiotherapy at the University of Alcalá de Henares and their Ethics Committee gave approval with number CEIM/HU/2019/50. All participants read the patient information sheet and their written consent was obtained before participation. The procedures with volunteers complied with the Declaration of Helsinki.

### 2.2. Sample Size

To calculate the sample size, the formula proposed by Zou et al., (2012) [27] was used with the average data of the three interobserver ICC measurements for the variable Diaphragm_TV_inspiration (see Table 1) of the first 20 subjects recruited, estimating a final sample of 52 subjects accepting an error α = 0.05 and a power of 80%.

### 2.3. Participants

A total of 54 participants were included in the study. The subjects should be between 18 and 60 years and agree to the design of the study, voluntarily participating in it and signing the informed consent. The sample was divided into two groups: case group (*n* = 23 subjects with NS-LBP) and control group (*n* = 31 subjects without NS-LBP). For inclusion in the case group, the characteristics of the NS-LBP must be: (1) low back pain located in the region between T12-L1 to L5, (2) unilateral or bilateral distribution within this region, (3) not irradiated to other regions, (4) not necessarily being continuous, and (5) an evolution of more than three months. For the control group, healthy subjects without NS-LBP at least during the previous three months or at present were included.

Exclusion criteria for both groups comprised the presence of congenital musculoskeletal disorders, spinal deformities, compression of nerve roots, oncological processes, or any other process that may be the cause of NS-LBP. The presence of NS-LBP that does not meet the characteristics described for inclusion were also considered as exclusion criteria. In addition, participants with body mass index (BMI) higher than 31 Kg/m^2^ and diagnoses of respiratory or neurological conditions were excluded.

### 2.4. Measurement Instruments and Examiners

A Vinno E35 ultrasound device (VINNO Technology, Suzhou, China) was used for all the ultrasonography measurements and a linear probe 7–18 Mhz, with 38 mm footprint, also a convex probe 1–5 Mhz was used with 52 mm footprint. A physiotherapy student carried out all the examinations. She received training in the measurement weeks prior to the start of the study. This training was given by a muscle skeletal sonographer expert in RUSI with at least 10 years of clinical experience. The sonographer was present during all the examinations.

### 2.5. Measures

Demographic and clinical information were collected. These variables included age, sex, height, weight, body mass index (BMI), physical activity level, and frequent sporting. It was also asked if they had NS-LBP and if so, pain intensity was measured with numeric pain rating scale (NRS Pain) [28].

The subjects were placed in a supine position (bed slope of 45°) with a roller cushion under the popliteal fossa to reduce tension on the lower limbs that could affect the measurements. The thickness of the diaphragm was measured over the area of apposition of the right hemidiaphragm with the linear transducer placed transversely over the eight or ninth intercostal space between the anterior axillary line and mamillar line [29] (Figure 1A). It was measured during tidal (TV) and force volume (FV) by B-mode ultrasound imaging. First, every subject was instructed to breathe quietly. Then, participants were instructed to inhale as much air as they could and release it as quickly and deeply as possible on demand. Three images were captured at the end of tidal inspiration and three more were taken at the end of tidal expiration in all the sample by two observers. The same number of images was done at force volume. The thickness of the diaphragm muscle was measured on frozen images by placing two calipers at the boundaries of the echogenic layer of muscle tissue [22] (Figure 1B).

We proceeded to take the diaphragmatic excursion. The convex transducer was placed longitudinally over the anterior subcostal region in a supine position (bed slope of 45°) [30] (Figure 2A). First, the right hemidiaphragmatic dome was identified as a thick hyperechogenic curved line and a caliper was placed over this line. The M-mode was activated to obtain three sinusoidal curves of three respiratory cycles at tidal breathing in all the sample by two observers. The subjects were then asked to perform three force breaths to obtain another three sinusoidal curves at force breathing. Measurement of diaphragmatic excursion was made on frozen images, placing a caliper at the maximum point of the curve, corresponding to the end of inspiration, and a second caliper at the minimum point of the curve, corresponding to the end of expiration [31] (Figure 2B).

The participants lay down in prone position with the headboard descended slightly lower than the plane of the stretcher and two cushions under the hip and the ankles. The arms were placed supported on the headboard at approximately 120° of shoulder abduction and 90° of elbow flexion [32]. The linear transducer was placed on the L4 vertebral level [33] (Figure 3A). The thickness of the multifid muscles was measured placing two calipers from the edge of the lamina of the vertebra to the hyperechogenic layer of the superficial fascia of the muscular belly at both rest and contraction [15]. For the activation of the multifidus muscles, participants were instructed to perform the contralateral arm lift (CAL) [34]. Three measurements were taken at both rest and contraction on frozen ultrasound images. Finally, the thickness of the TLF was measured on the frozen images of the lumbar multifid muscle at rest. For this, two calipers were placed between the limits of the hyperechogenic layer of the TLF [34] (Figure 3B).

### 2.6. Statistical Analysis

Statistical analysis was conducted using “R Ver. 3.5.1” (R Foundation for Statistical Computing, Institute for Statistics and Mathematics, Welthandelsplatz 1, 1020 Vienna, Austria). A P value of less than 0.05 was considered to be statistically significant. This showed a large number of variables with a non-normal distribution, which, together with the low sample size (*n* < 30), made it advisable to use non-parametric tests. The distribution of the variables in each group was tested with the Shapiro–Wilk test. Finally, the Mann–Whitney U test and permutation tests (Monte-Carlo simulation and exact test) were applied.

The outcome measures of this study were the thickness of the right hemidiaphragm at inspiration and expiration at tidal and force breathing, the diaphragmatic excursion at tidal and force breathing, the thickness of the right and left lumbar multifidus muscles at rest and contraction, and the thickness of the TLF. The diaphragm thickening capacity at tidal (TV) and force volume (FV) was also calculated by thickening fraction (TF): (thickness at end-inspiration − thickness at end-expiration)/thickness at end-expiration. The intra-rater reliability analysis was calculated for the thickness diaphragm variable [35] using two-way mixed single measures (Consistency/Absolute agreement) and the formula used for the MDC calculation was (SEM × √2 × 1.96), where SEM was (SD × √1 − ICC) [36]. 

## 3. Results

### Participants Demographic

Table 1 presents the demographic characteristics of the total sample. The case group consisted of 10 men and 13 women with a mean age of 25.13 ± 10.04 years. 65.2% of the subjects performed any one of the sports detailed in Table 1. The mean score in the NRS Pain was 5.57 ± 1.53, as well as 56.5% had pain in bilateral localization and 43.5% presented unilaterally.

The control group consisted of 12 men and 19 women with a mean age of 22.94 ± 5.23 years. 90.3% of the subjects performed any one of the sports detailed in Table 2.

The values of the different measures obtained by the two observers are presented in Table 3.

Mann–Whitney U test results indicated that there were no statistically significant differences between the activation of the diaphragm in relation to the lumbar multifidus musculature and TLF in subjects with and without NS-LBP. The results of the permutation test with Monte-Carlo simulation and exact test showed statistically significant differences in the variables: thickness of the diaphragm at tidal volume inspiration and expiration, thickness of the diaphragm at force volume inspiration, and TF at tidal and force breathing. Table 4 shows meaningful differences between both groups.

Finally, the intraclass correlation index (ICC) was calculated. There was a good intra-rater reliability, exceeding or within the limit of 0.8, except in the variable thickness of the diaphragm at tidal volume inspiration. The values of ICC are presented in Table 1.

## 4. Discussion

The main findings of the study showed that diaphragmatic activation does not correlate with the activity of the lumbar multifidus muscles and TLF in healthy subjects nor subjects with NS-LBP. However, subjects with NS-LBP showed higher diaphragm thickness in inspiration and expiration at tidal breathing, as well as higher diaphragm thickness in inspiration at force breathing. In addition, they showed a higher capacity of thickening of the diaphragm at tidal and force breathing expressed as TF.

The results of this study contrast with previous research conducted among athletes with and without lumbopelvic pain [8], which reported a reduced diaphragm thickness in athletes who suffered from lumbopelvic pain. This variability has been reported in a recent study of Haaksma et al. [37] reflecting that diaphragm thickness in the zone of apposition on the mid-axillary line has significant variability in the craniocaudal direction. According to these considerations, they recommended that measurements should be performed carefully and interpreted with caution due to the relevant variability in the thickness of the diaphragm in the mid-axillary line.

Regarding the excursion of the diaphragm, no differences were found between the two groups. Kolář et al. [10] also found no difference in diaphragm excursion in subjects with and without chronic low back pain by magnetic resonance imaging. However, the results of this study contrast with studies which reported a lower diaphragmatic excursion in subjects with structural alterations of the lumbar spine evaluated with dynamic magnetic resonance imaging [6] and a reduced diaphragmatic excursion in subjects with NS-LBP evaluated with B-mode ultrasound [9]. This variability in the outcome, as far as ultrasound is concerned, could not be attributed to the distinction in the method of evaluation of the diaphragmatic excursion with B-mode or M-mode ultrasound [37]. In fact, it has been demonstrated that RUSI technique is a valid method for muscles assessment of low back region compared to magnetic resonance [38,39].

The results of the measurement of lumbar multifidus muscles contrast with the studies where it was found that subjects with chronic low back pain showed a lower size of lumbar multifidus muscles [18] and a smaller size of the lumbar multifidus muscles ipsilateral compared to the symptoms in subjects with acute low back pain [19]. These findings could be because these authors measured the size of the lumbar multifidus muscles at various vertebral levels of the lumbar spine. However, our study was limited exclusively to measuring the thickness of the lumbar multifidus muscles at the L4 level. In addition, these authors excluded participants who had practiced some type of sport that could involve the lumbar multifidus muscles. In our research, due to the high percentage of sportsmen in both groups (see Table 1), sports practice could attenuate the difference in the thickness of this musculature.

Almazán-Polo et al. [40] also found a reduced difference in CSA of multifidus muscles in athletes with chronic lumbopelvic pain (CLPP) compared to healthy athletes. These findings could contrast with our study because these authors included subjects with bilateral lumbopelvic pain in the past two years and in our study subjects are included with NS-LBP (unilateral or bilateral) in the past three months.

No significant differences in TLF thickness were also found. These findings are in line with Almazán-Polo et al. [41] and contrast with the findings of Langevin et al. [14], which reported a thickness greater than 25% of the TLF in subjects with chronic low back pain. However, ultrasound images show more disorganized appearance of the connective layers that compose this structure in subjects with NS-LBP, according to Langevin et al. [14].

Our main hypothesis that diaphragm activity correlated with the activity of the lumbar multifidus muscles and TLF was rejected after the results of this investigation, though with a larger sample than in this pilot study, the hypothesis could be reconsidered. In addition, recent research was conducted on the effect of diaphragm training on lumbar stabilizer muscles [42] and they found an increase in the thickness measured by ultrasound not only of the diaphragm but also of the multifidus muscles. This suggests that both are functionally related and this would support our hypothesis about the correlation between both. However, more future studies are needed to establish a protocol that the patient should follow when being evaluated at a clinical setting.

In addition, our study obtained high values of intraobserver reliability, whose data are in line with the high values of intraobserver reliability obtained by an examiner trained by a sonographer weeks prior to the start of the study [22], or the findings of another study where a physiotherapist with minimal training achieved results with excellent reliability [23].

According to the SEM and MDC values (Table 4), these were lower than the statistically significant differences for the right hemi-diaphragm thickness at inspiration and expiration between both groups. This provides greater reliability data on the measurement results. To the best our knowledge, this is the first study to report that SEM and MDC values for the diaphragm ICC thickness contraction prove to be robust and reliable measures.

Today, we cannot explain the absence of correlation between these structures in healthy subjects due to the lack of previous studies to rely on to explain this. However, it is possible to answer the lack of relationship between these structures in subjects with NS-LBP. In this study, statistically significant differences were found between diaphragm thicknesses. Despite this, it did not correlate with activation of the lumbar multifidus muscles and TLF. So, it is deduced that the diaphragm excursion is the key variable in this correlation subject to study.

Kocjan et al. [10] hypothesized that decreased diaphragm excursion could lead to a decrease in IAP and minimize stress on TLF. In subjects with NS-LBP maintain normal diaphragmatic excursion values—as in our study—the PIA would maintain its common values and the tension over the TLF would not be modified. This aspect could explain the finding of no correlation between these structures.

The limitations of this study are that due to the global pandemic COVID-19 and the obligation to wear a mask during the taking of measurements, participants reported difficulties in achieving full air filling at forced volume which could modify the results in the measurements of thickness and diaphragmatic excursion at forced volume. The evaluator who assigned participants to both groups and performed the measurements was not blinded, so this might have affected the measurements.

## 5. Conclusions

Diaphragmatic activity was not correlated with lumbar multifidus muscles and TLF in subjects with and without NS-LBP. However, patients with NSL-BP showed higher diaphragm thickness in inspiration and expiration at tidal volume, and inspiration at force volume. In addition, patients with NS-LBP showed higher capacity of thickening of the diaphragm at tidal and force volume which will have to be analyzed in future studies. More thickness does not match with no pain and may be a complex muscle synergy of muscles where some muscles retract and others atrophy.

## Figures and Tables

**Figure 1 medicina-59-00315-f001:**
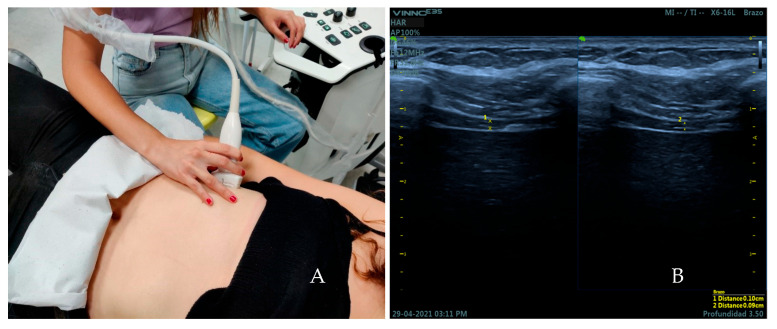
(**A**) Placement of the transducer over the area of apposition of the right hemidiaphragm for the measurement of the thickness of the diaphragm. (**B**) Ultrasound image obtained over this area. Distance 1 corresponds to the thickness of the diaphragm at the end of inspiration. Distance 2 corresponds to the thickness of the diaphragm at the end of expiration.

**Figure 2 medicina-59-00315-f002:**
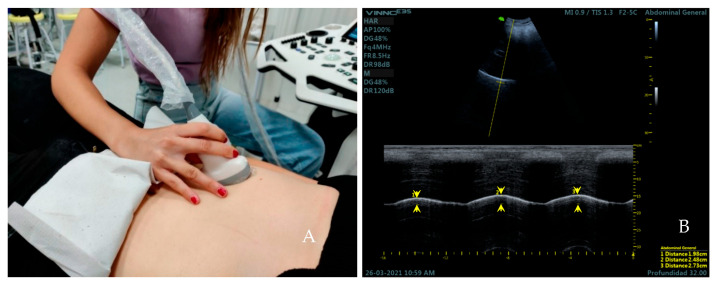
Placement of the transducer over the anterior subcostal region for the measurement of diaphragmatic excursion (**A**). Ultrasound image obtained over this area (**B**). Distance 1, 2, and 3 (yellow arrows) correspond to the excursion of the diaphragm during the first, second, and third respiratory cycle at current volume; respectively.

**Figure 3 medicina-59-00315-f003:**
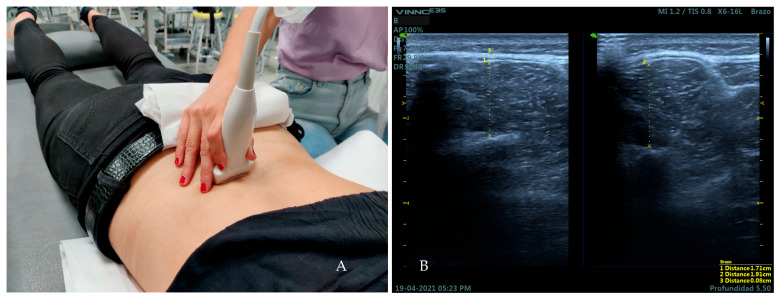
Placement of the transducer over the L4 vertebral level for the measurement of the variables of the lumbar multifidus muscles and TLF (**A**). Ultrasound image obtained over this area (**B**). Distance 1 corresponds to the thickness of the lumbar multifidus muscle at rest Distance 2 corresponds to the thickness of the lumbar multifidus muscle at contraction. Distance 3 corresponds to the thickness of the TLF.

**Table 1 medicina-59-00315-t001:** Intra-rater reliability of thickness of the diaphragm measurements.

	ICC 95%CI	Average Measurement	SEM 95%CI	MDC
Observer 1: Diaphragm TV inspiration	0.876 (0.813, 0.922)	0.158	0.02 (0.017, 0.024)	0.056
Observer 1: Diaphragm TV aspiration	0.818 (0.731, 0.884)	0.128	0.016 (0.014, 0.019)	0.046
Observer 1: Diaphragm FV inspiration	0.787 (0.688, 0.863)	0.284	0.048 (0.041, 0.055)	0.134
Observer 1: Diaphragm FV aspiration	0.759 (0.651, 0.844)	0.140	0.022 (0.018, 0.026)	0.061
Observer 2: Diaphragm TV inspiration	0.763 (0.656, 0.846)	0.176	0.024 (0.02, 0.029)	0.067
Observer 2: Diaphragm TV aspiration	0.788 (0.689, 0.864)	0.144	0.02 (0.016, 0.024)	0.055
Observer 2: Diaphragm FV inspiration	0.631 (0.489, 0.752)	0.303	0.06 (0.053, 0.068)	0.167
Observer 2: Diaphragm FV aspiration	0.763 (0.657, 0.847)	0.161	0.021 (0.018, 0.024)	0.058

Variables are in centimeters.

**Table 2 medicina-59-00315-t002:** Demographic characteristics of subjects. Data are reported ad the mean ± standard deviation or with absolute and relative values (%).

Variable		Case Group	Control Group
*n*		23	31
Age		25.13 ± 10.04	22.94 ± 5.23
Sex, *n*(%)	Male	10 (43.5)	12 (38.7)
	Female	13 (56.5)	19 (61.3)
BMI		22.11 ± 2.84	22.30 ± 2.00
NRS Pain		5.57 ± 1.53	No data
Pain location, *n*(%)	Bilateral	13 (56.5)	0 (0.0)
	Right side	6 (26.1)	0 (0.0)
	Left side	4 (17.4)	0 (0.0)

Data expressed as mean ± typical deviation with absolute and relative values (%).

**Table 3 medicina-59-00315-t003:** Measures of the diaphragm activity sampled by observers.

	Observer 1	Observer 2
*n*	52	52
Diaphragm TV inspiration measure 1	0.15 ± 0.05	0.17 ± 0.05
Diaphragm TV inspiration measure 2	0.16 ± 0.06	0.17 ± 0.05
Diaphragm TV inspiration measure 3	0.16 ± 0.06	0.18 ± 0.05
Diaphragm TV expiration measure 1	0.13 ± 0.04	0.14 ± 0.04
Diaphragm TV expiration measure 2	0.13 ± 0.04	0.14 ± 0.04
Diaphragm TV expiration measure 3	0.13 ± 0.04	0.15 ± 0.04
Diaphragm FV inspiration measure 1	0.28 ± 0.10	0.29 ± 0.09
Diaphragm FV inspiration measure 2	0.28 ± 0.11	0.31 ± 0.09
Diaphragm FV inspiration measure 3	0.28 ± 0.10	0.31 ± 0.11
Diaphragm FV expiration measure 1	0.14 ± 0.05	0.15 ± 0.04
Diaphragm FV expiration measure 2	0.14 ± 0.04	0.16 ± 0.04
Diaphragm FV expiration measure 3	0.14 ± 0.05	0.16 ± 0.04

Data expressed as mean ± standard deviation.

**Table 4 medicina-59-00315-t004:** Finals results of the study. Data are reported ad the mean ± standard deviation or with absolute and relative values (%).

	Case Group	Control Group	*p* ^a^	Difference (95%CI)	r (95%CI)
*n*	23	31			
Thickness of the diaphragm in inspiration at tidal breathing	0.17 ± 0.07	0.14 ± 0.04	<0.001	0.02 (0, 0.05)	0.26 (0.026, 0.477)
Thickness of the diaphragm in expiration at tidal breathing	0.14 ± 0.03	0.12 ± 0.04	<0.001	0.02 (0, 0.04)	0.242 (0.019, 0.535)
Thickness of the diaphragm in inspiration at force breathing	0.31 ± 0.11	0.26 ± 0.08	<0.001	0.04 (−0.01, 0.09)	0.207 (0.022, 0.454)
Thickness of the diaphragm in expiration at force breathing	0.14 ± 0.04	0.14 ± 0.04	1	0 (−0.02, 0.02)	0.034 (0.007, 0.279)
TF at tidal breathing	0.28 ± 0.29	0.20 ± 0.08	<0.001	0.024 (−0.031, 0.081)	0.136 (0.004, 0.374)
TF at force breathing	1.23 ± 0.65	0.97 ± 0.55	<0.001	0.236 (−0.047, 0.5)	0.217 (0.02, 0.449)
Diaphragmatic excursion at tidal breathing	1.94 ± 1.05	1.78 ± 0.76	1	0.08 (−0.27, 0.42)	0.072 (0.008, 0.36)
Diaphragmatic excursion at force breathing	4.50 ± 1.38	4.51 ± 1.09	1	−0.092 (−0.75, 0.65)	0.031 (0.002, 0.402)
Thickness of the left lumbar multifidus at contraction	2.99 ± 0.64	2.96 ± 0.74	1	0.02 (−0.37, 0.45)	0.015 (0.002, 0.288)
Thickness of the left lumbar multifidus at rest	2.69 ± 0.53	2.59 ± 0.62	1	0.1 (−0.26, 0.42)	0.082 (0.001, 0.359)
Thickness of the right lumbar multifidus at contraction	3.14 ± 0.64	3.03 ± 0.63	1	0.07 (−0.31, 0.54)	0.051 (0.002, 0.356)
Thickness of the right lumbar multifidus at rest	2.73 ± 0.55	2.66 ± 0.52	1	0.05 (−0.29, 0.39)	0.039 (0.003, 0.278)
Thickness of the TLF	0.18 ± 0.06	0.18 ± 0.06	1	0 (−0.04, 0.03)	0.024 (0.008, 0.287)

Data expressed as mean ± typical deviation with absolute and relative values (%). Variables are in centimeters. ^a^ significative if *p* < 0.05.

## Data Availability

The data presented in this study are available on request from the corresponding author.

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
