# Peer review of "Diaphragmatic Activation Correlated with Lumbar Multifidus Muscles and Thoracolumbar Fascia by B-Mode and M-Mode Ultrasonography in Subjects with and without Non-Specific Low Back Pain: A Pilot Study"

_medicina, 2023, doi:10.3390/medicina59020315_

Round 1

Reviewer 1 Report (Previous Reviewer 2)

Manuscript Number: medicina-2178657

Full Title: Diaphragmatic activation correlated with lumbar multifidus muscles and thoracolumbar fascia by B-mode and M-mode ultrasonography in subjects with and without non-specific low back pain: a pilot study.

Recommendation Rating: 75   

GENERAL COMMENT:

    This paper presents the evaluation of the functional connections of the diaphragm, the lumbar multifidus muscles and the thoracolumbar fascia in subjects with and without NS-LBP by means of a comparison based on ultrasonographic parameters.

The work appears to have been carried out carefully in places, and the limitation of correlation analysis and tested conditions of breathing maneuvers have been improved. The revised manuscript addressed and answered the previous questions. The preliminary results will be of considerable value to anyone trying to understand the connections of diaphragmatic activation, lumbar multifidus muscles and thoracolumbar fascia in low-back pain patients.

Author Response

Many thanks for the comments.

Reviewer 2 Report (Previous Reviewer 1)

Dear Authors,

Thank you very much for your changed. I recommed this manuscript to publish. 

Author Response

Many thanks for the comments.

This manuscript is a resubmission of an earlier submission. The following is a list of the peer review reports and author responses from that submission.

Round 1

Reviewer 1 Report

Dear Authors,

The manuscript titled as " Diaphragmatic activation correlated with lumbar multifidus muscles and thoracolumbar fascia by B-mode and M-mode ultrasonography in subjects with and without non-specific low back pain: a pilot study.". 

The paper is overall well structured and contains an important topic.  I have one suggestion that could improve the paper: Please add legend below table. The introduction explains the most important information about the study and the discussion is well written. 

Kind regards

Author Response

Dear reviewer. Thanks for the comment. The legend below tables has been included.

Reviewer 2 Report

Manuscript Number: medicina-1918598

Full Title: Diaphragmatic activation correlated with lumbar multifidus muscles and thoracolumbar fascia by B-mode and M-mode ultrasonography in subjects with and without non-specific low back pain: a pilot study.

Recommendation Rating: 60   

GENERAL COMMENT:

    This paper presents the evaluation of the functional connections of the diaphragm, the lumbar multifidus muscles and the thoracolumbar fascia in subjects with and without NS-LBP by means of a comparison based on ultrasonographic parameters.

The work appears to have been carried out carefully but, in places, the limitation of correlation analysis and tested conditions of breathing maneuvers could lead to the result explanation is not clear enough. Therefore, the conclusion that the diaphragmatic activation had no functional correlation with the activation of lumbar multifidus muscles and TLF for both groups may not be scientific. The preliminary results will be of considerable value to anyone trying to understand the connections of diaphragmatic activation, lumbar multifidus muscles and thoracolumbar fascia in low-back pain patients.

SPECIFIC COMMENTS:

Abstract

    The manuscript investigated on assessing Diaphragmatic activation, however, no significant differences recorded between the activation of the diaphragm and the activation of the lumbar multifidus muscles and TLF were found. The diaphragm thickening capacity was also calculated by thickening fraction (TF) at tidal volume and force volume. The research hypotheses were presented at this moment and suggest to descript and confine to the tested conditions.

Introduction:

The paragraph about “The diaphragm raises the intra-abdominal pressure prior to the start of a functional task with the upper limbs and keeps it at high levels while the task is being carried out.”, however, the measurement of different volume of breathing pattern in the present study may not be the similar fashion. The connection between arm tasks and regular breathing behaviors need to be reviewed to fit the proposed research subjects.

Methods:

1. Please clarify the reason for using the Mann-Whitney U test and permutation tests for the investigation of relationship of diaphragmatic activation, lumbar multifidus muscles and thoracolumbar fascia in low-back pain patients. Why the ANOVA or the t-tests are not selected for the comparisons between two different subject groups.

2. The diaphragm thickening capacity at tidal and force volume was also calculated by thickening fraction (TF): (thickness at end-inspiration – thickness at end-expiration)/thickness at end-expiration. The above measurement in the results and discussion were not well-explained in the later part of the manuscript.

Results

1. There are a lot of typo error in “Table 1 and Table 2. Demographic caracteristics of subjects. Data are reported ad the mean ± standar desviation or with absolute and relative values (%).”

2. The selection of the statistical analysis in absolute and relative values (%) is not explained in the paper.  The P value (Pa) with a footnote is not explained.

Discussion:

1. The paragraph 1 in the discussion: The main findings of the study showed that diaphragmatic activation does not correlate with the activity of the lumbar multifidus muscles and TLF in healthy subjects nor subjects with NS-LBP. It is inappropriate to address the final interpretation. The major concerns about this study is the tested conditions. The regular breathing conditions in the tidal respiration and forced respiration may not be necessary to strongly recruit the lumbar multifidus muscles and thoracolumbar fascia.  

2. Please also check the paper about diaphragm training to expand the discussion:

J Pain Res . 2018 Nov 28;11: 3031-3045. doi: 10.2147/JPR.S181610. The effect of diaphragm training on lumbar stabilizer muscles: a new concept for improving segmental stability in the case of low back pain

Author Response

GENERAL COMMENT:

    This paper presents the evaluation of the functional connections of the diaphragm, the lumbar multifidus muscles and the thoracolumbar fascia in subjects with and without NS-LBP by means of a comparison based on ultrasonographic parameters.

The work appears to have been carried out carefully but, in places, the limitation of correlation analysis and tested conditions of breathing maneuvers could lead to the result explanation is not clear enough. Therefore, the conclusion that the diaphragmatic activation had no functional correlation with the activation of lumbar multifidus muscles and TLF for both groups may not be scientific. The preliminary results will be of considerable value to anyone trying to understand the connections of diaphragmatic activation, lumbar multifidus muscles and thoracolumbar fascia in low-back pain patients.

Dear reviewer. Thanks for the comment. Currently, the scientific evidence in relation to the core musculature and the presence of low back pain presented an inverse correlation; however, in this study, although changes in the morphology of the musculature were found, no significant differences were found when comparing the degree of activation measured ultrasonographically between the groups.

SPECIFIC COMMENTS:

Abstract

    The manuscript investigated on assessing Diaphragmatic activation, however, no significant differences recorded between the activation of the diaphragm and the activation of the lumbar multifidus muscles and TLF were found. The diaphragm thickening capacity was also calculated by thickening fraction (TF) at tidal volume and force volume. The research hypotheses were presented at this moment and suggest to descript and confine to the tested conditions.

Dear reviewer. Thanks for the comment. To evaluate by RUSI (Rehabilitative Ultrasound Imaging) the muscular thickness at rest of the abdominal wall, the excursion of the pelvic floor and the respiratory diaphragm, as well as to study their activity through specific activation maneuvers previously validated for each of them and under the same conditions for all participants.

Introduction:

The paragraph about “The diaphragm raises the intra-abdominal pressure prior to the start of a functional task with the upper limbs and keeps it at high levels while the task is being carried out.”, however, the measurement of different volume of breathing pattern in the present study may not be the similar fashion. The connection between arm tasks and regular breathing behaviors need to be reviewed to fit the proposed research subjects.

Dear reviewer. Thanks for the comment.  For the present investigation we proposed to evaluate the degree of muscular contraction of each one of the muscular groups according to the maneuvers that are described in the literature, and that have been shown to be the ones with the greatest validity and reliability[1–5]. (see references at the end)

Methods:

  1. Please clarify the reason for using the Mann-Whitney U test and permutation tests for the investigation of relationship of diaphragmatic activation, lumbar multifidus muscles and thoracolumbar fascia in low-back pain patients. Why the ANOVA or the t-tests are not selected for the comparisons between two different subject groups.

Dear reviewer. Thanks for the comment. The distribution of the variables in each group was tested with the Shapiro-Wilk test, which showed a large number of variables with a non-normal distribution, which, together with the low sample size (n<30 in each group) made it advisable to use non-parametric tests, specifically the Mann-Whitney U test and permutation tests, both with Monte-Carlo and exact simulation.

  1. The diaphragm thickening capacity at tidal and force volume was also calculated by thickening fraction (TF): (thickness at end-inspiration – thickness at end-expiration)/thickness at end-expiration. The above measurement in the results and discussion were not well-explained in the later part of the manuscript.

 Dear reviewer. Thanks for the comment. The results about the diaphragm thickening were commented at the beginning of the discussion and in the new discussion redaction. 

Results

  1. There are a lot of typo error in “Table 1 and Table 2. Demographic caracteristics of subjects. Data are reported ad the mean ± standar desviation or with absolute and relative values (%).”

Dear reviewer. Thanks for the comment. Tables 1 and 2 has been fixed.

  1. The selection of the statistical analysis in absolute and relative values (%) is not explained in the paper.  The P value (Pa)with a footnote is not explained.

 Dear reviewer. Thanks for the comment. The footnotes has been added and explained.

Discussion:

  1. The paragraph 1 in the discussion: The main findings of the study showed that diaphragmatic activation does not correlate with the activity of the lumbar multifidus muscles and TLF in healthy subjects nor subjects with NS-LBP. It is inappropriate to address the final interpretation. The major concerns about this study is the tested conditions. The regular breathing conditions in the tidal respiration and forced respiration may not be necessary to strongly recruit the lumbar multifidus muscles and thoracolumbar fascia.

    Dear reviewer. Thanks for the comment. The point commented by the reviewer looks have sense to be considering the maximum recruitment, but the conditions how the test were done with the validated activities has reported these results. In this context the results showed no correlation in controverse with the widely accepted hypothesis the diaphragmatic and lumbar activity correlation.

  1. Please also check the paper about diaphragm training to expand the discussion:

J Pain Res . 2018 Nov 28;11: 3031-3045. doi: 10.2147/JPR.S181610. The effect of diaphragm training on lumbar stabilizer muscles: a new concept for improving segmental stability in the case of low back pain

Dear reviewer. Thanks for the comment. The recommended paper has been reviewed, considered for discussion and included. The discussion has been enhanced and enlarged.

REFERENCES

  1. Sweeney N, O’Sullivan C, Kelly G. Multifidus muscle size and percentage thickness changes among patients with unilateral chronic low back pain (CLBP) and healthy controls in prone and standing. Man Ther [Internet]. 2014 Oct 1;19:433–9.
  2. Stokes M, Rankin G, Newham DJ. Ultrasound imaging of lumbar multifidus muscle: normal reference ranges for measurements and practical guidance on the technique. Man Ther [Internet]. 2005 Jan 1;10:116–26.
  3. Cohn D, Benditt JO, Eveloff S, McCool FD. Diaphragm thickening during inspiration. J Appl Physiol. 1997;83(1):291–6. doi:10.1152/jappl.1997.83.1.291
  4. Sarwal A, Walker FO, Cartwright MS. Sarwal A, Walker FO, Cartwright MS. Neuromuscular ultrasound for evaluation of the diaphragm. Muscle Nerve 2013; 47: 319-329. Muscle Nerve. 2013;4(164):319–29. doi:10.1002/mus.23671.Neuromuscular
  5. Testa A, Soldati G, Giannuzzi R, Berardi S, Portale G, Gentiloni Silveri N. Ultrasound M-Mode Assessment of Diaphragmatic Kinetics by Anterior Transverse Scanning in Healthy Subjects. Ultrasound Med Biol [Internet]. 2011 Jan [cited 2019 May 1];37(1):44–52. doi:10.1016/j.ultrasmedbio.2010.10.004